# Isorhamnetin Reduces Glucose Level, Inflammation, and Oxidative Stress in High-Fat Diet/Streptozotocin Diabetic Mice Model

**DOI:** 10.3390/molecules28020502

**Published:** 2023-01-04

**Authors:** Abdelrahim Alqudah, Esam Y. Qnais, Mohammed A. Wedyan, Sara Altaber, Yousra Bseiso, Muna Oqal, Rawan AbuDalo, Khaled Alrosan, Amjad Z. Alrosan, Suhad Bani Melhim, Mohammad Alqudah, Rabaa Y. Athamneh, Omar Gammouh

**Affiliations:** 1Department of Clinical Pharmacy and Pharmacy Practice, Faculty of Pharmaceutical Sciences, The Hashemite University, Zarqa 13133, Jordan; 2Department of Biology and Biotechnology, Faculty of Science, The Hashemite University, Zarqa 13133, Jordan; 3Department of Pharmaceutical Technology, Faculty of Pharmaceutical Sciences, The Hashemite University, Zarqa 13133, Jordan; 4Department of Physiology, College of Medicine and Medical Sciences, Arabian Gulf University, Manama 329, Bahrain; 5Department of Physiology and Biochemistry, College of Medicine, Jordan University of Science and Technology, Irbid 22110, Jordan; 6Department of Medical Laboratory Sciences, Faculty of Allied Science, Zarqa University, Zarqa 13133, Jordan; 7Department of Clinical Pharmacy and Pharmacy Practice, Faculty of Pharmacy, Yarmouk University, Irbid 21163, Jordan

**Keywords:** isorhamnetin, insulin resistance, diabetes, oxidative stress, inflammation

## Abstract

Background: Isorhamnetin is a flavonoid that is found in medical plants. Several studies showed that isorhamnetin has anti-inflammatory and anti-obesity effects. This study aims to investigate the anti-diabetic effects of isorhamnetin in a high-fat diet and Streptozotocin-(HFD/STZ)-induced mice model of type 2 diabetes. Materials and Methods: Mice were fed with HFD followed by two consecutive low doses of STZ (40 mg/kg). HFD/STZ diabetic mice were treated orally with isorhamnetin (10 mg/kg) or (200 mg/kg) metformin for 10 days before sacrificing the mice and collecting plasma and soleus muscle for further analysis. Results: Isorhamnetin reduced the elevated levels of serum glucose compared to the vehicle control group (*p* < 0.001). Isorhamnetin abrogated the increase in serum insulin in the treated diabetic group compared to the vehicle control mice (*p* < 0.001). The homeostasis model assessment of insulin resistance (HOMA-IR) was decreased in diabetic mice treated with isorhamnetin compared to the vehicle controls. Fasting glucose level was significantly lower in diabetic mice treated with isorhamnetin during the intraperitoneal glucose tolerance test (IPGTT) (*p* < 0.001). The skeletal muscle protein contents of GLUT4 and p-AMPK-α were upregulated following treatment with isorhamnetin (*p* > 0.01). LDL, triglyceride, and cholesterol were reduced in diabetic mice treated with isorhamnetin compared to vehicle control (*p* < 0.001). Isorhamnetin reduced MDA, and IL-6 levels (*p* < 0.001), increased GSH levels (*p* < 0.001), and reduced GSSG levels (*p* < 0.05) in diabetic mice compared to vehicle control. Conclusions: Isorhamnetin ameliorates insulin resistance, oxidative stress, and inflammation. Isorhamnetin could represent a promising therapeutic agent to treat T2D.

## 1. Introduction

Diabetes is considered one of the most common metabolic disorders worldwide. According to the International Diabetes Federation (IDF), around 537 million adults are living with diabetes, and this number is projected to rise to 643 million in 2030 [1]. Diabetes is characterized by hyperglycemia, affecting carbohydrate, fat, and protein metabolism resulting from abnormal insulin secretion, insulin resistance, or both factors. Type 2 diabetes (T2D) is the most prevalent type of diabetes which comprises around 90% of all diabetes cases in the world [2,3]. T2D is characterized by insulin resistance associated with relative insulin deficiency that generally occurs in adults 40 years old or above who are obese or overweight [4].

Skeletal muscles are the major site of insulin-mediated glucose uptake, and insulin-sensitive glucose transporter-4 (GLUT4) mediates this uptake [5]. The activation of AMP-activated protein kinase (AMPK), which increases the translocation of GLUT4 to the cell surface, is another insulin-independent mechanism that facilitates glucose uptake in skeletal muscle [6,7]. Therefore, the activation of the AMPK-GLUT4 pathway is an effective way to improve insulin sensitivity in T2D.

Additionally, oxidative stress and inflammation have a part in the etiology of T2D and associated consequences [8]. Type 1 and type 2 diabetes have been linked to pancreatic cell destruction as a result of inflammation and oxidative stress [9]. The development of insulin resistance and T2D, as well as diabetes complications, were consistently linked to elevated levels of pro-inflammatory mediators in obese individuals [10,11], suggesting that inflammation and oxidative stress play a vital role in T2D development and complications.

O-methylated flavonol, isorhamnetin, is found commonly in medical plants such as *Oenanthe javanica*, *Hippophae rhamnoides*, and *Ginkgo biloba* [12]. It acts as an anti-inflammatory by decreasing the inflammatory cell, and proinflammatory gene expression and inhibits NF-kB luciferase activity [13]. Furthermore, studies showed that isorhamnetin has anti-obesity and anti-diabetic effects [14]. In addition, this flavonoid reduced oxidative stress and hyperglycemia in the streptozotocin (STZ)-induced diabetic model [15]. Moreover, isorhamnetin has improved insulin secretion and the expression of enzymes that are involved in lipid metabolism in mouse-induced obesity model [16]. Additionally, isorhamnetin was shown to promote carbohydrate metabolism by inhibiting carbohydrate hydrolyzing enzymes such as α-amylase and α-glucosidase in vitro which leads to a slowdown in carbohydrate digestion, thus prolonging the time for carbohydrate digestion causing a reduction in glucose absorption, in addition to alleviation of insulin secretion impairment [17]. Moreover, in the non-alcoholic steatohepatitis (NASH) mouse model, isorhamnetin was able to improve glucose uptake in the liver and muscles [18]. Furthermore, in silico study showed that isorhamnetin can interact with the active site of GLUT4 protein through an H-bond interaction [19]. Consistent with this finding, Eid et al. demonstrated that isorhamnetin could stimulate AMPK and increase GLUT4 translocation in rat L6 skeletal muscle cells [20]. These results show that isorhamnetin has anti-diabetic benefits in different models, but more pre-clinical research is needed to determine its applicability in the treatment of type 2 diabetes. Therefore, the objectives of this work were to examine how isorhamnetin affected the AMPK-GLUT4 pathway in a high-fat diet (HFD)/STZ-induced mouse model.

## 2. Results

### 2.1. The Hypoglycemic Effect of Isorhamnetin

Serum glucose was significantly higher in the vehicle control diabetic group compared to the non-diabetic group (Figure 1A, *n* = 6, *p* < 0.001). Treatment with isorhamnetin significantly reduced the serum glucose levels compared to the vehicle control group in the presence of T2D (Figure 1A, *n* = 6, *p* < 0.001). Similarly, the serum glucose level in the diabetic group was significantly reduced with metformin treatment compared to the vehicle control (Figure 1A, *n* = 6, *p* < 0.001). Unexpectedly, insulin levels were significantly increased in the vehicle control diabetic group compared to the non-diabetic group (Figure 1B, *n* = 6, *p* < 0.001); however, treating diabetic mice with isorhamnetin significantly reduced insulin levels compared to the vehicle control group (Figure 1B, *n* = 6, *p* < 0.001). Treating diabetic mice with metformin significantly reduced insulin levels compared to the vehicle control group (Figure 1B, *n* = 6, *p* < 0.001). No difference was observed between isorhamnetin and metformin in terms of glucose and insulin.

To study the effect of isorhamnetin on insulin resistance, HOMA-IR was measured. The presence of T2D was confirmed by HOMA-IR, which was significantly increased in the vehicle control diabetic group compared to the non-diabetic group (Figure 1C, *n* = 6, *p* < 0.001). Interestingly, isorhamnetin was able to restore HOMA-IR in the diabetic group, which was comparable to the non-diabetic group. The same effect on HOMA-IR was observed when diabetic mice were treated with metformin. HOMA-IR was not different between isorhamnetin and metformin groups. Moreover, blood glucose level during IPGTT was significantly lower in isorhamnetin and metformin groups compared to vehicle control (Figure 2, *n* = 6, *p* < 0.001). Furthermore, water and food consumption was higher in diabetic groups compared to the non-diabetic group (Figure 3A,B, respectively). No difference was found between the isorhamnetin and metformin groups and the vehicle control group in water and food consumption (Figure 3A,B, respectively).

To determine the mechanism by which isorhamnetin improves blood glucose and insulin resistance, GLUT4 and p-AMPK-α protein expression in skeletal muscle tissue were measured. GLUT4 protein expression was significantly downregulated in the presence of T2D (Figure 4A, *n* = 6, *p* < 0.05); however, treating diabetic mice with isorhamnetin significantly upregulated GLUT4 expression compared to the vehicle control diabetic group (Figure 4A, *n* = 6, *p* < 0.01). Metformin also was able to upregulate GLUT4 expression compared to the vehicle control diabetic group (Figure 4A, *n* = 6, *p* < 0.001). Similarly, p-AMPK-α expression was significantly downregulated as a result of T2D (Figure 4B, *n* = 6, *p* < 0.001), and treating diabetic mice with either isorhamnetin or metformin significantly upregulated p-AMPK-α expression compared to the vehicle control diabetic group (Figure 4B, *n* = 6, *p* < 0.01, <0.05, respectively). No difference was observed in GLUT4 and p-AMPK-α expression between isorhamnetin and metformin groups.

### 2.2. The Effect of Isorhamnetin on Lipid Profile

As depicted by Figure 5, dyslipidemia was clearly present in diabetic mice. Triglyceride (Figure 5A, *n* = 6, *p* < 0.001), total cholesterol (Figure 5B, *n* = 6, *p* < 0.001), and LDL (TGs; Figure 5C, *n* = 6, *p* < 0.001) were significantly higher in the vehicle control diabetic group compared to the non-diabetic group. Treating diabetic mice with isorhamnetin significantly reduced serum triglyceride (Figure 5A, *n* = 6, *p* < 0.001), total cholesterol (Figure 5B, *n* = 6, *p* < 0.01), and LDL (Figure 5C, *n* = 6, *p* < 0.001) levels compared to the vehicle control diabetic group. Similarly, metformin was able to reduce triglyceride (Figure 5A, *n* = 6, *p* < 0.05), cholesterol (Figure 5B, *n* = 6, *p* < 0.01), and LDL (Figure 5C, *n* = 6, *p* < 0.001) levels significantly in diabetic mice in comparison to the vehicle controls.

### 2.3. The Effect of Isorhamnetin on GSH, GSSG, MDA and IL-6

Serum GSH expression was significantly reduced in vehicle control diabetic mice compared to non-diabetic mice (Figure 6A, *n* = 6, *p* < 0.001); however, treating diabetic mice with either isorhamnetin or metformin demonstrated a significant increase in GSH compared to the vehicle control diabetic group (Figure 6A, *n* = 6, *p* < 0.05, 0.01, respectively). Moreover, serum GSSG was significantly increased in vehicle control diabetic mice compared to non-diabetic mice (Figure 6B, *n* = 6, *p* < 0.05), however, isorhamnetin and metformin were able to reduce GSSG levels in diabetic mice compared to vehicle control (Figure 6B, *n* = 6, *p* < 0.05). On the other hand, serum MDA (Figure 6C, *n* = 6, *p* < 0.001) and IL-6 (Figure 6D, *n* = 6, *p* < 0.001) concentrations were significantly increased in the presence of T2D. Interestingly, isorhamnetin showed an ability to reduce serum MDA and IL-6 levels significantly in diabetic mice in comparison to the vehicle control group (Figure 6C,D, *n* = 6, *p* < 0.001). The same effect was observed when diabetic mice were treated with metformin (Figure 6C,D, *n* = 6, *p* < 0.001).

## 3. Discussion

T2D is characterized mainly by high blood glucose associated with insulin resistance [21] which can cause several complications including cardiovascular disease, nephropathy, retinopathy, and neuropathy [22]. In our study, the results demonstrated that isorhamnetin significantly reduced glucose levels, which was comparable to the gold standard, metformin, suggesting that isorhamnetin represents a promising emerging hypoglycemic agent. Furthermore, HFD/STZ combination resembles the characteristics of T2D such as early stage hyperinsulinemia, hyperglycemia, hyperlipidemia, and β-cells dysfunction [23]. Furthermore, our results reported increased systemic insulin in the vehicle control group which is aligned with several studies reporting that hyperinsulinemia exists, since the peripheral tissues lack their insulin-sensitizing property that ultimately results in hyperglycemia, which leads to an increase in insulin secretion in the early stages of T2D as part of the compensatory mechanism that aims to counteract the presence of insulin resistance in T2D [24,25,26,27]. Oddly enough, isorhamnetin was effective at bringing blood insulin levels back to normal or non-diabetic levels. When metformin was administered to diabetic mice, the same result was seen. Isorhamnetin enhanced insulin sensitivity in diabetic mice, and this is in tandem with the decrease in serum insulin observed following treatment with isorhamnetin. HOMA-IR was calculated to further understand the effect of isorhamnetin on insulin resistance.

Previous studies established that HFD, which is enriched by saturated fatty acids, impaired cellular glucose uptake and induced insulin resistance [28]. Notably, saturated fatty acids enhance lipids accumulation in muscles, thereby inducing insulin resistance [29]. Palmitate, for example, as a saturated fatty acid, promotes cytokines secretion as IL-6 and TNF-α that can lead to insulin resistance and glucose intolerance [30]. Meanwhile, HFD downregulates the expression of GLUT4, which induces glucose intolerance [31]. It has been reported that the activated AMP protein kinase (AMPK) plays a significant role in regulating cellular energy metabolism. Its malfunction is associated with insulin resistance and other metabolic disorders. Metformin alters the AMP/ATP ratio which activates AMPK through phosphorylation which improves glucose utilization [32]. However, HFD is attributed to a decrease in the phosphorylation of AMPK, thus reducing glucose uptake. Metformin has been shown to enhance the expression of GLUT4 and activation of AMPK through phosphorylation, thus increasing glucose uptake by cells. The isorhamnetin mechanism of action in reducing hyperglycemia could be similar to metformin which needs to be investigated further. The mammalian target of rapamycin (mTOR) is a serine and threonine protein kinase that has an established role in insulin resistance and AMPK directly phosphorylates Raptor, which is a component of mTORC1, to repress mTORC1 [33]. A recent study showed that isorhamnetin decreased the expression of mTOR [25] which might be also another mechanism by which isorhamnetin improves insulin sensitivity which needs to be studied in such a model of diabetes. In order to investigate the hypoglycemic mode of action of isorhamnetin, the p-AMPK and GLUT4 levels in skeletal muscle were measured. About 45–50% of the body’s mass is made up of skeletal muscle, which also transports 80% of the body’s glucose [34]. In skeletal muscles, AMPK regulates the transcription of the GLUT-4 gene. GLUT-4 is a critical glucose transporter, transporting extracellular glucose to insulin-sensitive cells to keep blood glucose homeostasis [35]. Moreover, reduced GLUT-4 skeletal muscle protein expression and inhibition of GLUT4 translocation results in insufficient glucose transportation and hence insulin resistance. Activation of the AMPK–GLUT4 pathway enhances insulin sensitivity which improves glucose control in T2D [36]. In addition, the role of AMPK in the prevention of T2D has previously been investigated in combination with the regulation of insulin signaling and GLUT-4 activity [37]. Our results demonstrate that isorhamnetin is capable of increasing the expression of both p-AMPK and GLUT4 in skeletal muscle suggesting that isorhamnetin can improve glucose uptake through the AMPK-GLUT4 pathway.

On the other hand, in addition to hyperglycemia, T2D is associated with dyslipidemia. High postprandial TGs, total cholesterol, and LDL define diabetic dyslipidemia [38]. These lipid alterations are the key factors leading to T2D-associated complications [39]. Particularly, dyslipidemia is a significant risk factor for macrovascular diabetes complications, and numerous studies have linked dyslipidemia to microvascular complications associated with T2D, such as diabetic retinopathy, diabetic nephropathy, and diabetic neuropathy [38]. Our findings revealed that isorhamnetin reduced TGs, total cholesterol, and LDL in our T2D model suggesting that isorhamnetin may reduce cardiovascular disease risk associated with T2D. This should be explored in future studies.

Many clinical and experimental studies show that there is a strong link between oxidative stress and the development of T2D, and its complication [40]. Oxidative stress is defined as reduced tolerance between oxidants and antioxidants, due to the production of reactive oxygen species (ROS) and reduction in the rate of antioxidant defense mechanisms including GSH (non-enzymatic antioxidant) [41]. ROS can damage the lipids, causing lipid peroxidation such as lipid peroxidation of low-density lipoprotein (oxLDL) or peroxidation of polyunsaturated fatty acid (oxPUFAs). Additionally, ROS induces the release of MDA, a highly reactive compound that interacts with protein, and nucleic acid, and causes damage to various tissues and cells [42]. MDA has been used as a biomarker of lipid peroxidation and as an indication of free radical damage in the blood [43]. Our findings show that isorhamnetin considerably reduces plasma MDA levels and increases GSH levels, implying that isorhamnetin could be effective as an antioxidant agent in T2D by reducing lipid peroxidation or increasing free radical scavenging activity.

IL-6 has complex and often conflicting activities. It promotes an anti-inflammatory (M2-like) state in macrophages. Consistent with these observations, others have reported that IL-6 functions to limit atheroma formation and that it is secreted in response to physical exercise, mediating its insulin-sensitizing actions [44,45]. On the other hand, IL-6 also acts as a pro-inflammatory cytokine involved in the acute phase reaction to tissue injury. It has a contributory role in a number of inflammatory and autoimmune diseases, and its secretion by the adipose tissues of obese organisms contributes to metabolic dysfunction including insulin resistance and promoting atherosclerosis [46,47]. Subclinical chronic inflammation has been implicated as an independent risk factor for the development and progression of T2D and its complications [8]. In particular, the multifunctional cytokine interleukin 6 (IL-6) has been linked to the pathogenesis of T2D. Increased levels of systemic IL-6 are a strong predictor of T2D and are thought to have a role in the development of inflammation, insulin resistance, and β-cell dysfunction [48]. In addition, mounting data shows that IL-6 impairs insulin signaling in hepatocytes, and inhibits glucose-stimulated insulin release from the pancreatic β-cell [48]. Moreover, many studies suggest that anti-inflammatory activity plays a key role in the prevention of T2D development and the reduction of the incidence of diabetes complications [48]. As stated in the introduction, inflammation and oxidative stress play a key role in the development of T2D and its complications. Therefore, reducing inflammation and oxidative stress will improve the outcomes in T2D. A large body of evidence showed that activation of AMPK reduces inflammation and oxidative stress via different mechanisms which have a protective effect in diabetes [49,50,51]. Our findings in this study showed that isorhamnetin has anti-inflammatory and antioxidant activities which are in HFD/STZ-induced diabetes model aligned with previous reports [13,52,53]. Suggesting that AMPK activation by isorhamnetin could reduce inflammation and oxidative stress in such a model.

In summary, isorhamnetin has the ability to reduce serum glucose levels and normalize insulin in the diabetic group compared to the vehicle control group. Additionally, the fasting glucose level was lower in the isorhamnetin group after IPGTT. In addition, isorhamnetin reduced HOMA-IR value in diabetic mice compared to vehicle control. These effects could be explained by the upregulation of p-AMPK-α levels which leads to an increase in the translocation of GLUT4 to the cell surface which promotes glucose uptake in the skeletal muscles, thus improving insulin sensitivity. Furthermore, isorhamnetin reduced LDL, cholesterol, and triglyceride levels in diabetic mice compared to vehicle control which might be through the activation of AMPK that is shown to promote glucose and fatty acid catabolism and prevents protein and fatty acid synthesis [54]. The reduction in LDL, cholesterol, and triglyceride could be a key contributor to reducing insulin resistance which might explain the improvement of insulin sensitivity observed in our study [55,56]. Moreover, isorhamnetin reduced MDA, GSSG, and IL-6 levels and increased GSH levels which suggests that isorhamnetin could have antioxidant and anti-inflammatory activities in T2D. These findings are in line with previous studies demonstrated that isorhamnetin reduced MDA levels and increased intracellular GSH levels by enhancing the activity of antioxidant enzymes, superoxide dismutase (SOD), and catalase (CAT) in STZ-induced type 1 diabetes rat model [57]. Another study also reported that isorhamnetin inhibited the NF-κB signaling pathway, which led to reduced levels of inflammatory mediators such as IL-6, ICAM-1, and TNF-α [52,58]. Taken together, the antioxidant and the anti-inflammatory activities of isorhamnetin could be through the activation of antioxidant enzymes and inhibiting NF-κB signaling pathway, which needs to be studied more in the future.

Limitations of this study include the following aspects: (i) isorhamnetin was administered for a short period of time (ii) GLUT4 expression was assessed using immunoblotting reflective of its total amount, however immunohistochemistry may be a better technique to assess its activity and translocation to the cell membrane, (iii) we only measured GLUT4 expression in the skeletal muscles, this should also be performed using liver and adipose tissue, and (iv) the anti-inflammatory and antioxidant activities of isorhamnetin should be studied more in T2D. Nevertheless, our findings in this study indicate the crucial role of isorhamnetin in improving typical features of T2D, which is the first report to date.

## 4. Materials and Methods

### 4.1. Induction of T2D and Experimental Design

Six-week-old male C57BL/6 mice were maintained under standard conditions including 12 h light/dark cycles and at 22 ± 2° temperature [59]. T2D was induced by feeding the experimental mice with HFD (60% fat, D14292, Research Diets, Inc., New Brunswick, NJ, USA) for 9 weeks followed by intraperitoneal injection of STZ (40 mg/kg). At week 10, mice were administered another intraperitoneal low dose of STZ (40 mg/kg) to complete the induction of diabetes. The HFD/STZ-induced diabetes model is a well-established model for diabetes in which HFD feeding will lead to obesity, hyperinsulinemia, and altered glucose homeostasis due to insufficient compensation by the beta cells of the pancreatic islets. A single high dose of STZ causes sudden and significant destruction of pancreatic cells; however, progressive multiple low doses of STZ after HFD as the model of this study causes less destruction of pancreatic cells which portraits the same characteristics and mimics the pathogenesis and clinical features of T2D in human [26,60]. One week after STZ injection, plasma glucose was measured and mice with a plasma glucose concentration of >200 mg/dL were considered to have developed T2D and selected for the subsequent experiments.

Mice were randomly divided into four groups (*n* = 6 each) as follows: (i) the normal control group (non-diabetic, ND) received a normal diet, (ii) the vehicle control (VC) diabetic group treated with dimethyl sulfoxide (DMSO, Panreac Quimica SA, Barcelona, Spain) only, (iii) diabetic group treated with 10 mg/kg isorhamnetin (Sigma-Aldrich, Hamburg, Germany), and (iv) diabetic group treated with 200 mg/kg Metformin (MeRCK, Frankfurt, Germany). The dose of 10 mg/kg isorhamnetin treatment was chosen based on previous study where they studied the effect of isorhamnetin on GLUT4 levels in HFD-induced obesity mice model [61]. In this study, they administered isorhamnetin in three different concentrations (10, 100, 1000 mg/kg) for 90 min. Isorhamnetin was able to significantly upregulate GLUT4 at doses of 10 and 100 mg/kg. Therefore, in our study, the lowest dose (10 mg/kg) was chosen to avoid the toxic effect which might occur with longer period of treatment. All treatments were given orally once per day. After 10 days of treatment, mice were fasted overnight (16 h) and then sacrificed using CO_2_ chamber, blood and skeletal muscle (soleus muscle) were collected for ex vivo analysis. Mice sacrificing after 10 days of treatment with isorhamnetin was based on previous studies where isorhamnetin treatment was performed for 10 days in T1D STZ-induced model and other models for different diseases [62,63,64].

### 4.2. Biochemical Investigations

#### 4.2.1. Measurement of Serum Glucose, Insulin, and Lipids

Serum glucose was determined using a commercial kit (Glucose assay kit, MyBioSource, San Diego, CA, USA). Serum insulin was measured by ELISA using commercial kit (mouse insulin ELISA kit, MyBioSource, USA). Triglyceride (TG, triglyceride assay kit), low-density lipoprotein (LDL, LDL assay kit), and cholesterol (total cholesterol assay kit) were determined using commercially available kits (MyBioSource, USA) according to the manufacturer’s instructions.

#### 4.2.2. Homeostasis Model Assessment of Insulin Resistance (HOMA-IR)

This model represents the interaction between fasting plasma insulin and fasting plasma glucose which is a useful tool for determining insulin resistance. According to international diabetes federation, the HOMA-IR cut-off level in healthy individuals is less than 1 and in men with diabetes is 1.55, and women with diabetes is 2.22 [65]. Studies showed that normal HOMA-IR in healthy mice is 1.9 and in STZ mice is 21.6 [66].

In the current study, we used the following formula to compute HOMA-IR:

HOMA-IR = (Fasting glucose (mg/dL) × Fasting insulin (μIU/mL)/405 [67]

The constant 405 is a normalizing factor representing the result of multiplication of the normal fasting plasma insulin level (μIU/mL) with the normal fasting plasma level (81 mg/dL) [68].

#### 4.2.3. Intraperitoneal Glucose Tolerance Test

Mice were given an intraperitoneal injection of glucose (0.5 g/kg) after being fasted for 18 h. Using a glucometer, blood glucose levels were measured from the tail vein at 0, 30, 60, and 120 min (Accu-Check Performa, Roche Diagnostics, Basel, Switzerland).

#### 4.2.4. Measurement of Serum Reduced Glutathione (GSH), Oxidized Glutathione (GSSG), Malondialdehyde (MDA), and IL-6 Levels

Reduced glutathione (GSH, GSH assay kit), oxidized glutathione (GSSG, GSSG assay kit), and IL-6 (IL-6 ELISA kit) levels were measured in the serum using commercially available kits (MyBioSource, USA). Plasma MDA level was determined by using commercial Thiobarbituric acid (TBA) Assay Kit (MyBioSource, USA) according to the manufacturer’s instructions.

### 4.3. Western Blotting

Skeletal muscle tissues (soleus muscle) were homogenized in radioimmunoprecipitation (RIPA)-lysis buffer, containing a protease inhibitor cocktail (Santa Cruze Biotechnology, Dallas, TX, USA) using a tissue homogenizer. Homogenates were centrifuged at 12,000× *g* for 20 min at 4 °C and supernatant was collected. The total protein was quantified using bicinchoninic acid assay kit (Bioquochem, Asturias, Spain). Equal amount of protein was separated by sodium dodecyl sulfate-polyacrylamide gel and then transferred to a nitrocellulose membrane (Thermo Fisher Scientific, Waltham, MA, USA). The membrane was blocked for 1 h at room temperature using 3% bovine serum albumin (BSA) before incubating overnight with either phosphorylated AMPK-α1 (p-AMPK-α1, Abcam, Cambridge, UK) or GLUT4 (MyBioSource, USA) primary antibodies (1:1000 dilution). The membrane was washed three times with washing buffer (Tween-20/Tris-buffered saline) before incubating it with the goat anti-rabbit secondary antibody (MyBioSource, USA, 1:5000 dilution) for 1 h at room temperature. Following incubation, the membrane was washed three times before submerging into the ECL substrate (ThermoScientific, USA) for one minute followed by imaging with chemiLITE Chemiluminescence Imaging System (Cleaver Scientific, Rugby, UK). To ensure equal protein gel loading, β-actin was used as a housekeeping gene (MyBioSource, USA, 1:10,000 dilution). The intensity of the bands was measured using Image J software and adjusted to β-actin.

### 4.4. Statistical Analysis

All analyzed parameters were tested for the normality of the data using Kolmogorov–Smirnov test. Data are represented as mean ± SEM. Differences between groups were calculated using one-way analysis of variance (ANOVA) followed by Tukey post hoc using Graphpad Prism software version (9.3.1). The significance value of difference was considered when the *p* value < 0.05.

## 5. Conclusions

In conclusion, our results demonstrated that isorhamnetin could be a very useful hypoglycemic agent for the treatment of T2D due to its multifactorial effects including (i) the reduction in insulin resistance, (ii) an increase in glucose uptake by the skeletal muscle, (iii) improvement in the lipid profile, (iv) reduction in oxidative stress and inflammation and (v) the activation of the GLUT4-AMPK pathway. The effects and mechanisms demonstrated by isorhamnetin were very similar to metformin.

## Figures and Tables

**Figure 1 molecules-28-00502-f001:**
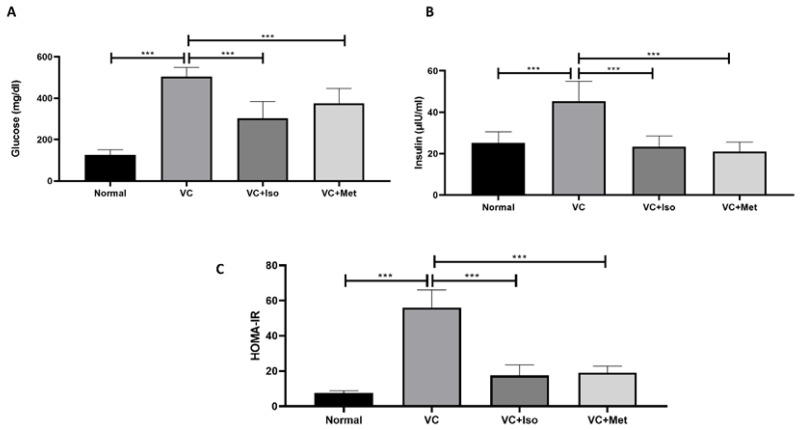
The anti-diabetic effect of isorhamnetin. Isorhamnetin significantly reduced glucose (**A**) and insulin (**B**) levels in diabetic mice. HOMA-IR (**C**) was significantly reduced with isorhamnetin treatment. Mice were fed with HFD for 8 weeks followed by two low doses of STZ injection (40 mg/kg) after diabetes confirmed, mice were treated with 10 mg/kg isorhamnetin or 200 mg/kg metformin for 10 days, mice were then sacrificed, and serum collected for ELISA analysis. One-way ANOVA followed by Tukey post hoc, *** *p* < 0.001. VC; vehicle control, Iso; isorhamnetin, Met; metformin.

**Figure 2 molecules-28-00502-f002:**
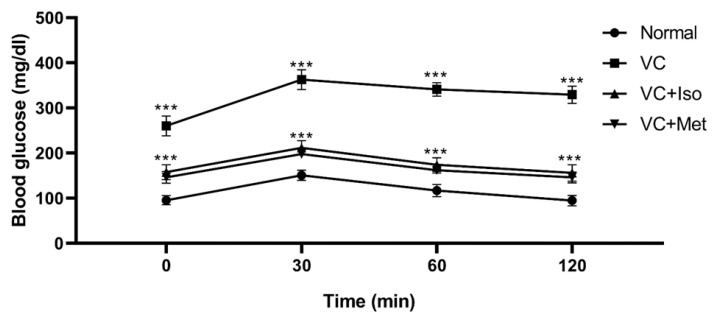
Isorhamnetin reduced glucose level during intraperitoneal glucose tolerance test (IPGTT). Isorhamnetin significantly reduced glucose in diabetic mice during intraperitoneal glucose tolerance test. Mice were fed with HFD for 8 weeks followed by two low doses of STZ injection (40 mg/kg), after diabetes confirmed, mice were treated with 10 mg/kg isorhamnetin or 200 mg/kg metformin for 10 days, mice were then fasted overnight before injection with 0.5 g/kg glucose intraperitoneally, and glucose level determined at 0, 30, 60, and 120 min. Two-way ANOVA followed by Tukey post hoc, *** *p* < 0.001. VC; vehicle control, Iso; isorhamnetin, Met; metformin.

**Figure 3 molecules-28-00502-f003:**
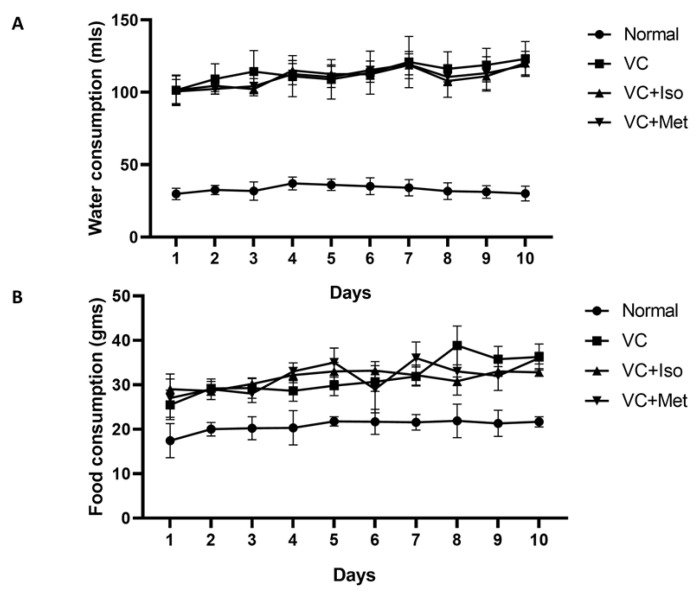
Water and food consumption was higher in diabetic groups. Water consumption (**A**) was significantly higher in diabetic groups compared to non-diabetic group. Food consumption (**B**) was higher in diabetic groups compared to non-diabetic group. Two-way ANOVA followed by Tukey post hoc. VC; vehicle control, Iso; isorhamnetin, Met; metformin.

**Figure 4 molecules-28-00502-f004:**
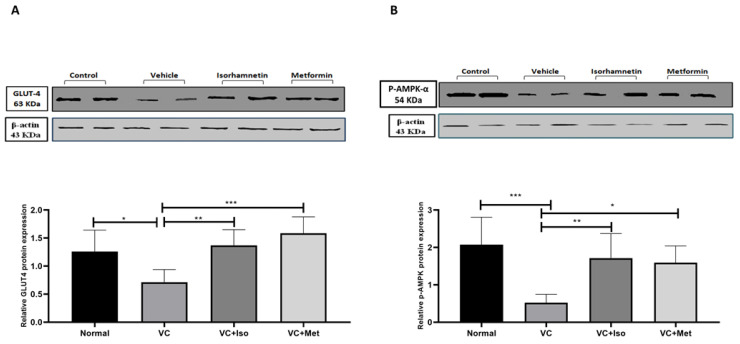
Isorhamnetin upregulated GLUT4 and p-AMPK-α expression in soleus muscle. Isorhamnetin significantly upregulated GLUT4 (**A**), and AMPK (**B**) expression in diabetic mice. Mice were fed with HFD for 8 weeks followed by two low doses of STZ injection (40 mg/kg), after diabetes confirmed, mice were treated with 10 mg/kg isorhamnetin or 200 mg/kg metformin for 10 days, mice were then sacrificed, and soleus muscle was isolated and homogenized before western blotting performed. One-way ANOVA followed by Tukey post hoc, * *p* < 0.05, ** *p* < 0.01, *** *p* < 0.001. VC; vehicle control, Iso; isorhamnetin, Met; metformin.

**Figure 5 molecules-28-00502-f005:**
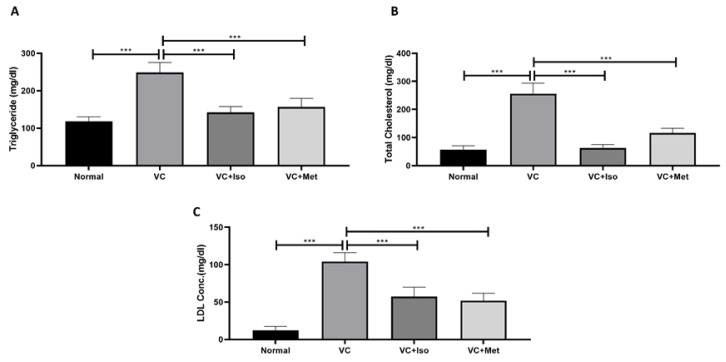
Isorhamnetin improves lipid profile in diabetes. Isorhamnetin significantly reduced triglyceride (**A**), cholesterol (**B**), and LDL (**C**) levels in diabetic mice. Mice were fed with HFD for 8 weeks followed by two low doses of STZ injection (40 mg/kg) after diabetes confirmed, mice were treated with 10 mg/kg isorhamnetin or 200 mg/kg metformin for 10 days, mice were then sacrificed, and serum collected for ELISA analysis. One-way ANOVA followed by Tukey post hoc, *** *p* < 0.001. VC; vehicle control, Iso; isorhamnetin, Met; metformin.

**Figure 6 molecules-28-00502-f006:**
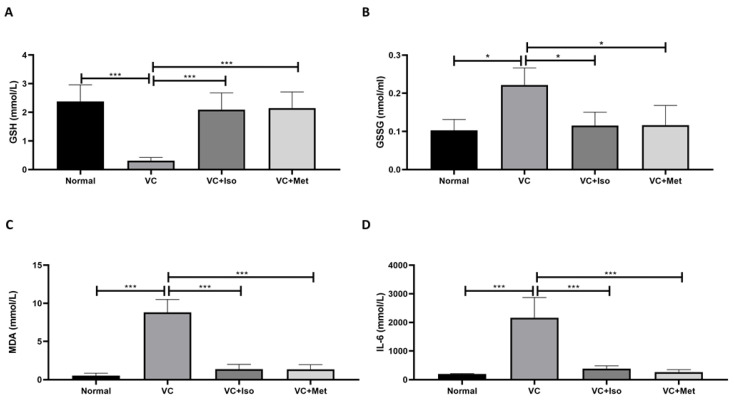
The anti-inflammatory effect of isorhamnetin. Isorhamnetin significantly increased GSH (**A**), and reduced GSSG (**B**), MDA (**C**), and IL-6 (**D**) levels in diabetic mice. Mice were fed with HFD for 8 weeks followed by two low doses of STZ injection (40 mg/kg) after diabetes confirmed, mice were treated with 10 mg/kg isorhamnetin or 200 mg/kg metformin for 10 days, mice were then sacrificed, and serum collected for ELISA analysis. One-way ANOVA followed by Tukey post hoc, * *p* < 0.05, *** *p* < 0.001. VC; vehicle control, Iso; isorhamnetin, Met; metformin.

## Data Availability

Open Science Framework. Isorhamnetin reduces glucose level, inflammation, and oxidative stress in high-fat diet/streptozotocin diabetic mice model. DOI: https://doi.org/10.17605/OSF.IO/CZ97A. The project contains the analysis file and original blots. The data are available under the terms of the Creative Common Attribution 4.0 International license (CC-BY 4.0).

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
