# Peer review of "Isorhamnetin Reduces Glucose Level, Inflammation, and Oxidative Stress in High-Fat Diet/Streptozotocin Diabetic Mice Model"

_molecules, 2023, doi:10.3390/molecules28020502_

Round 1

Reviewer 1 Report

In this study, the team investigated the anti-diabetic effects of isorhamnetin in a high-fat diet and Streptozotocin-(HFD/STZ)-induced mice model of type 2 diabetes.

They showed that Isorhamnetin reduced the serum glucose, abrogated the serum insulin in the treated diabetic group compared to the non-treated-diabetic mice. LDL, triglyceride, and cholesterol were reduced in diabetic mice treated with isorhamnetin compared to diabetic control. Also, skeletal muscle protein contents of GLUT4 and AMPK were upregulated following treatment with isorhamnetin. They concluded that Isorhamnetin could represent a promising therapeutic agent to treat T2D.

The study in general is very interesting and well-presented and written.

I do have 2 comments:

1-     What is the rationale for sacrificing the mice after 10 days of treatment?

2-     The mice were injected with 10 mg isorhamnetin. Have they tried different concentrations or based on what concentration was selected?

3-     WB images is not satisfactory; a better image is required and labeled with the molecular weight of the GLUT4 or AMPK.

Author Response

Dear editor and reviewer

We would like to thank you for the invested time and effort in carefully reviewing our manuscript. We are grateful for giving us the opportunity to revise our manuscript. Your comments were very useful and helped us in improving our manuscript. After careful consideration of the comments, the revision included many positive changes as suggested.

Changes are indicated in the tracked changes mode.

In this study, the team investigated the anti-diabetic effects of isorhamnetin in a high-fat diet and Streptozotocin-(HFD/STZ)-induced mice model of type 2 diabetes.

They showed that Isorhamnetin reduced the serum glucose, abrogated the serum insulin in the treated diabetic group compared to the non-treated-diabetic mice. LDL, triglyceride, and cholesterol were reduced in diabetic mice treated with isorhamnetin compared to diabetic control. Also, skeletal muscle protein contents of GLUT4 and AMPK were upregulated following treatment with isorhamnetin. They concluded that Isorhamnetin could represent a promising therapeutic agent to treat T2D.

C: The study in general is very interesting and well-presented and written.

R: The authors would like to thank you for your positive feedback.

I do have 2 comments:

  • What is the rationale for sacrificing the mice after 10 days of treatment?

R1: Thank you for your comment. The rationale for sacrificing the mice after 10 days of treatment with isorhamnetin was based on previous studies where isorhamnetin treatment was performed for 10 days in T1D STZ-induced model and other models for different diseases [1–3]. This now is added to the manuscript in the methods section.

2-     The mice were injected with 10 mg isorhamnetin. Have they tried different concentrations or based on what concentration was selected?

 R2: Thank you for you comment. The dose of 10 mg/kg isorhamnetin treatment was chosen based on previous study where they studied the effect of isorhamnetin on GLUT4 levels in HFD-induced obesity mice model [4]. In this study, they administered isorhamnetin in three different concentrations (10, 100, 1000 mg/kg) for 90 min. Isorhamnetin was able to significantly upregulate GLUT4 at doses of 10 and 100 mg/kg. Therefore, in our study the lowest dose (10 mg/kg) was chosen to avoid the toxic effect which might occur with longer period of treatment. This now is added to the manuscript in the methodology section.

  • WB images is not satisfactory; a better image is required and labeled with the molecular weight of the GLUT4 or AMPK.

R3: Thank you for your comment. The authors apologize for providing non-satisfactory WB images. Better images are added now to the figure.

Reviewer 2 Report

IPGTT in Abstract should be defined. Indicate also that the activation de AMPK was measured (p-AMPK).

Levels of GSSG should be measured.

NF kappa-B no beta.

If anti-inflammatory and antioxidant activities were already reported, authors should clearly establish what are the principal findings of this research. In this context, the last sentence in Discussion can be controversial and should be deleted.

How authors explain the increased insulin levels by the vehicle?

What are the HOMA-IR normal values in healthy mice and in STZ-model? In the HOMA-IR calculi what is the meaning of the constant of 450 in the formula? What are the normal ranges of HOMA-IR in healthy and diabetic patients and in murine models?

Authors stablish that they are determining the mechanism of action with their study, and they also mention that studied the action modo of isorhamnetin. However, they do not indicate the receptor(s) with the that this compound interacts. Please clarify. The data does not demonstrated mechanisms of action. How isorhamnetin can alter these parameters? With what receptors the isorhamnetin interact? In the same context, what are the mechanisms of metformin?

If the mechanism of the diabetogenic agent used implicate the elimination of the beta-pancreatic cells, authors should clearly stablish in the manuscript how the streptozotocin-induced diabetes model and fat can cause hyperinsulinemia. This model should be considered chemically induced, better than T2D. Kind of HFD 60% fat? Include references for the induction diabetes experimental model.

The modifications in IL-6 levels not always can be associated with pro-inflammatory effects. In general, IL-6 is considered a pleiotropic cytokine that in some conditions also can present beneficial effects. Therefore, the anti-inflammatory effect potential attributed to isorhamnetin should be better supported.  The same for antioxidant activity: GSSG should be measured also. Antioxidant activity and anti-inflammatory activity were not suitably demonstrated, and more experimental evidence should be given.

Minor:

First paragraph pag. 8: Mice.

4.2.2. Homeostasis.

Authors should indicate the time of fasting given in experimental animals.

Use better g units than rpm for centrifugation.

The format of the references should be uniformed.

Author Response

Dear editor and reviewer

We would like to thank you for the invested time and effort in carefully reviewing our manuscript. We are grateful for giving us the opportunity to revise our manuscript. Your comments were very useful and helped us in improving our manuscript. After careful consideration of the comments, the revision included many positive changes as suggested.

Changes are indicated in the tracked changes mode.

C1: IPGTT in Abstract should be defined. Indicate also that the activation de AMPK was measured (p-AMPK).

R1: Thank you for your comment. IPGTT is defined now in the abstract and p-AMPK is indicated

C2: Levels of GSSG should be measured.

R2: The authors agree with your suggestion and GSSG levels were measured now and included in the manuscript.

C3: NF kappa-B no beta.

R3: Apology for this mistake. It is corrected now.

C4: If anti-inflammatory and antioxidant activities were already reported, authors should clearly establish what are the principal findings of this research. In this context, the last sentence in Discussion can be controversial and should be deleted.

R4: Thank you for your comment. Apology for not establishing the principal findings of this study. The main finding of this study is that isorhamnetin has the ability to activate AMPK-GLUT4 pathway and reduced the associated inflammation and oxidative stress. This now amended in the objectives of the study and the following is added to the discussion.

’As stated in the introduction that inflammation and oxidative stress play a key role in the development of T2D and its complications. Therefore, reducing inflammation and oxidative stress will improve the outcomes in T2D. A large body of evidence showed that activation of AMPK reduces inflammation and oxidative stress via different mechanisms which have a protective effect in diabetes [1–3]. Our findings in this study showed that isorhamnetin has anti-inflammatory and antioxidant activities which is in HFD/STZ-induced diabetes model aligned with previous reports [4–6]. Suggesting that AMPK activation by isorhamnetin could reduce inflammation and oxidative stress in such model.’’  

C5: How authors explain the increased insulin levels by the vehicle?

R5: Thank you for your comment. The following is added now to the discussion.

‘’ Furthermore, HFD/STZ combination resembles the characteristics of T2D such as early-stage hyperinsulinemia, hyperglycaemia, hyperlipidaemia and β-cells dysfunction [7]. Several studies reported that hyperinsulinemia exists since the peripheral tissues lack their insulin sensitizing property that’s ultimately results in hyperglycaemia which lead to increase in insulin secretion in the early stages of T2D as part of the compensatory mechanism that aims to counteract the presence of insulin resistance in T2D [8–10].’’

C6: What are the HOMA-IR normal values in healthy mice and in STZ-model?

R6: Thank you for your comment. Studies showed that normal HOMA-IR in healthy mice is 1.9 and in STZ mice is 21.6 [11]

C7:  In the HOMA-IR calculi what is the meaning of the constant of 450 in the formula?

R7: Thank you for your comment. The constant 405 is a normalizing factor representing the result of multiplication of the normal fasting plasma insulin level (μU/mL) with the normal fasting plasma level (81 mg/dl) [12]

C8: What are the normal ranges of HOMA-IR in healthy and diabetic patients and in murine models?

R8: Thank you for your comment. According to international diabetes federation, the HOMA-IR cut-off level in healthy individuals is less than 1 and in men with diabetes is 1.55 and women with diabetes is 2.22 [13]

C9: Authors stablish that they are determining the mechanism of action with their study, and they also mention that studied the action modo of isorhamnetin. However, they do not indicate the receptor(s) with the that this compound interacts. Please clarify. The data does not demonstrated mechanisms of action. How isorhamnetin can alter these parameters? With what receptors the isorhamnetin interact? In the same context, what are the mechanisms of metformin?

R9: Thank you for your comment. The following is added now to the discussion to explain the mechanism by which isorhamnetin attenuates hyperglycemia.

’Previous studies established that HFD which is enriched by saturated fatty acids, impaired cellular glucose uptake and induced insulin resistance [14]. Notably, saturated fatty acids enhance lipids accumulation in muscles, thereby, induce insulin resistance [15]. Palmitate for example as a saturated fatty acid, promotes cytokines secretion as IL-6 and TNF-α that can lead to insulin resistance and glucose intolerance [16]. Meanwhile, HFD downregulates the expression of GLUT4 which induces glucose intolerance [17]. It has been reported that the activated AMP- protein kinase (AMPK) plays a significant role to regulate cellular energy metabolism.  Its malfunction is associated with insulin resistance and other metabolic disorders. Metformin alter AMP/ATP ratio which activates AMPK through phosphorylation which improves glucose utilization [18]. However, HFD attributed to decrease the phosphorylation of AMPK, thus, reduces glucose uptake. Metformin have shown to enhance the expression of GLUT4 and activation of AMPK through phosphorylation, thus increases glucose uptake by cells. Isorhamnetin mechanism of action in reducing hyperglycaemia could be similar to metformin that needs to be investigated further. Mammalian target of rapamycin (mTOR) is a serine and threonine protein kinase that has an established role in insulin resistance and AMPK directly phosphorylates Raptor, which is a component of mTORC1, to repress mTORC1 [19].A recent study showed that isorhamnetin decreased the expression of mTOR [9] which might be also another mechanism by which isorhamnetin improves insulin sensitivity which needs to be studied in such model of diabetes.’’

C10: If the mechanism of the diabetogenic agent used implicate the elimination of the beta-pancreatic cells, authors should clearly stablish in the manuscript how the streptozotocin-induced diabetes model and fat can cause hyperinsulinemia. This model should be considered chemically induced, better than T2D. Kind of HFD 60% fat? Include references for the induction diabetes experimental model.

R10: Thank you for your comment. The following is added now to the methods section to describe the HFD/STZ-induced diabetes model with references. Also, the kind of 60% fat diet is added also in the methods section. 

’The HFD/STZ-induced diabetes model is a well-established model for diabetes in which HFD feeding will lead to obesity, hyperinsulinemia and altered glucose homeostasis due to insufficient compensation by the beta cells of the pancreatic islets. A single high dose of STZ causes sudden and significant destruction of pancreatic cells, however, progressive multiple low doses of STZ after HFD as the model of this study causes less destruction of pancreatic cells which portraits the same characteristics and mimics the pathogenesis and clinical features of T2D in human [10,20].

C11: The modifications in IL-6 levels not always can be associated with pro-inflammatory effects. In general, IL-6 is considered a pleiotropic cytokine that in some conditions also can present beneficial effects. Therefore, the anti-inflammatory effect potential attributed to isorhamnetin should be better supported.  The same for antioxidant activity: GSSG should be measured also. Antioxidant activity and anti-inflammatory activity were not suitably demonstrated, and more experimental evidence should be given.

R11: Thank you for your comment and apology for not mentioning the beneficial effects of IL-6. The following is added now to the discussion to clarify the beneficial effects of IL-6. Also, we agree that the anti-inflammatory and antioxidant activities of isorhamnetin should be studied more, however, more experiments that were supposed to be performed were not done due to limited funds which is mentioned now in the limitations of this study. Meanwhile, the GSSG is measured and added to the results.

‘’ IL-6 has complex and often conflicting activities. It promotes an anti-inflammatory (M2-like) state in macrophages. Consistent with these observations, others have reported that IL-6 functions to limit atheroma formation and that it is secreted in response to physical exercise, mediating its insulin-sensitizing actions [21,22]. On the other hand, IL-6 also acts as a pro-inflammatory cytokine involved in the acute phase reaction to tissue injury. It has a contributory role in a number of inflammatory and autoimmune diseases, and its secretion by the adipose tissues of obese organisms contributes to metabolic dysfunction including insulin resistance and promoting atherosclerosis [23,24]’’

Minor:

C1: First paragraph pag. 8: Mice.

R1: Corrected.

C2: 4.2.2. Homeostasis.

R2: Corrected.

C3: Authors should indicate the time of fasting given in experimental animals.

R3: Thank you for your comments. The fasting time is now added to the methodology section.

C4: Use better g units than rpm for centrifugation.

R4: Thank you for this comment. Corrected

C5: The format of the references should be uniformed.

R5: Thank you for your comment. The references are revised now and uniformed.

Round 2

Reviewer 1 Report

I am happy with the revised version.

Author Response

C: I am happy with the revised version.

R: Thank you very much.